# Genetic control of dynamic brain network reconfiguration during working memory

Maryam Fatemi, Mohammad Reza Daliri*

Neuroscience & Neuroengineering Research Lab., Biomedical Engineering Department, School of Electrical Engineering, Iran University of Science & Technology (IUST), Narmak, Tehran, Iran

* daliri@iust.ac.ir

## Abstract

Working memory is fundamental to human cognition, yet the genetic contributions to dynamic brain network states during task remain poorly understood. Here, we examined static and dynamic functional connectivity in monozygotic (MZ) and dizygotic (DZ) twins performing a 2-back versus 0-back working memory task, isolating cognitive-load-specific neural processes. Static functional connectivity exhibited moderate heritability (A ≈ 0.30), predominantly in heteromodal association cortex, including the Default Mode Network (DMN) and fronto-cingulate control regions, whereas primary sensory-motor networks showed minimal genetic influence. Dynamic connectivity further distinguished mean and variance components: mean dynamic connectivity demonstrated moderate heritability (A ≈ 0.23–0.30) across DMN–attention and DMN–frontal control interactions, whereas variance of dynamic connectivity showed the strongest genetic effects (A ≈ 0.25–0.44), reflecting highly heritable moment-to-moment neural flexibility within temporal, cingulo-opercular, and executive networks. Using k-means clustering, we identified two recurring dynamic brain states. State 2 was dominant (occupancy 75.6%) and stable (mean dwell ≈ 3.98 windows), whereas State 1 was transient (occupancy 24.4%, dwell ≈ 1.29 windows). Notably, neither state occupancy nor switching dynamics showed significant heritability, indicating these temporal characteristics are largely driven by environmental or task-related variability. Collectively, these findings reveal a gradient of genetic influence: static connectivity captures a moderately heritable trait-like baseline; mean dynamic connectivity reflects intermediate task-modulated coordination; and dynamic variability represents the most genetically influenced phenotype, highlighting neural flexibility and adaptability as heritable traits. These results suggest that variability in dynamic connectivity may serve as a sensitive endophenotype for cognitive and psychiatric traits, emphasizing the importance of temporal network dynamics in capturing genetically mediated individual differences in working memory.

**Data availability statement:** All the data used are from public databases (https://human-connectome.org/study/hcp-young-adult) and described and referenced properly in the manuscript.

**Funding:** The author(s) received no specific funding for this work.

**Competing interests:** The authors have declared that no competing interests exist.

# 1. Introduction

Working memory (WM) is a fundamental cognitive function that supports decision-making and executive control by temporarily maintaining and manipulating information [1]. When performing WM tasks, activity levels fluctuate across different brain regions, and the strength and direction of functional interactions between these regions also vary across individuals [2]. A key question in cognitive neuroscience is to what extent these inter-individual differences in functional brain connectivity are influenced by genetic versus environmental factors.

Twin studies and imaging genetics offer a powerful framework to address this question. By comparing monozygotic (MZ) twins (who share 100% of their genes) with dizygotic (DZ) twins (who share, on average, 50% of their segregating genes), researchers can estimate the heritability of brain function—that is, the proportion of variance in functional connectivity that can be attributed to genetic influences. Such investigations can identify endophenotypes linked to cognitive abilities and neuropsychiatric disorders.

While genetic influences on brain structure have been extensively studied [3–9], there has been comparatively less focus on the genetic basis of functional networks, especially during task performance [10–23].Most heritability studies of functional connectivity have focused on the resting-state, reporting moderate heritability estimates (15–42%) for key networks such as the default mode network (DMN), frontoparietal, and salience networks [10,12,17,19,22,24]. However, heritability estimates in task-based functional connectivity remain underexplored. Task-based studies suggest that genetic influences may vary depending on cognitive demand and the brain networks involved [15–18]. Yet, no study has systematically investigated the heritability of functional connectivity during an N-back working memory task, leaving a critical gap in our understanding of how genes influence dynamic brain function during cognition. Functional connectivity during cognitive tasks may be governed by different genetic and environmental influences than at rest. Understanding these task-specific genetic effects is essential for characterizing the heritability of cognitive functions.

Furthermore, most prior studies have examined static functional connectivity, which provides a summary measure of connectivity across an entire scan but fails to capture moment-to-moment fluctuations in connectivity. However, emerging evidence suggests that these dynamic variations in connectivity may be highly informative and relevant for cognitive flexibility, attention, and neuropsychiatric conditions [25–29]. Despite this, the heritability of dynamic functional connectivity remains largely unexplored—especially during a working memory task.

Dynamic functional connectivity has been increasingly recognized for its role in distinguishing between healthy individuals and clinical populations. For example, studies have shown that dynamic functional connectivity measures are more sensitive than static measures in predicting schizophrenia symptoms [30], capturing momentary fluctuations in attention and cognitive states [31], and detecting brain network disruptions in ADHD [32].The variance of dynamic connectivity (i.e., the extent to which connectivity fluctuates over time) has been found to differentiate patients from controls in conditions like schizophrenia and ADHD [32,33]. However, no study

has yet examined the heritability of dynamic variance during a working memory task—a crucial step in determining whether these dynamic properties are genetically influenced and could serve as potential endophenotypes for cognitive function and mental health.

To address these gaps, we leverage the Human Connectome Project (HCP) dataset, which includes a large number of twin pairs, allowing us to estimate the heritability of both static and dynamic functional connectivity during a working memory task (2-back condition). Specifically, we Estimate the heritability of static functional connectivity between key working memory regions. Then we Investigate the genetic influences on dynamic functional connectivity, focusing on both the mean connectivity and the variance of dynamic fluctuations over time and comparing the heritability of static and dynamic measures to assess whether dynamic connectivity carries stronger genetic signatures than static connectivity.

Based on previous findings in resting-state functional connectivity studies, we hypothesize that Functional connectivity during the working memory task will exhibit moderate heritability (~15–40%), with the strongest genetic influences in regions known to be involved in WM (e.g., prefrontal and parietal cortices) and dynamic variance in connectivity may show stronger heritability than static connectivity, suggesting a genetic contribution to the flexibility of functional networks.

In this study, we use the Glasser parcellation to better capture functionally defined cortical regions. By applying genetic modeling, we aim to provide new insights into how genetic factors shape the stability and flexibility of functional connectivity during working memory processing.

Using data from the Human Connectome Project (HCP), we analyze fMRI data from 134 monozygotic (MZ) and 78 dizygotic (DZ) twins performing an N-back working memory task. By applying ACE genetic modeling, we estimate the heritability of static connectivity, mean dynamic functional connectivity (dFC), and variance of dFC across brain regions defined by the Glasser atlas. Our findings will provide novel insights into the genetic basis of working memory network dynamics, bridging gaps in imaging genetics research.

## 2. Methods and materials

### Data description

In this study, the HCP1200 Human Connectome dataset was used https://humanconnectome.org/study/hcp-young-adult.

We confirm that all methods were carried out in accordance with relevant guidelines and regulations. Data from the Human Connectome Project (HCP) were collected with the approval of the Washington University Institutional Review Board (IRB). Our study adhered to the HCP data usage agreement, ensuring compliance with ethical and regulatory standards. All participants provided informed consent in compliance with the Declaration of Helsinki and Informed consent was obtained from all participants and/or their legal guardians for participation in the study. Additionally, specific consent was secured for the publication of any identifying information and/or images in an online open-access format. Any data utilized adhered to the regulations stipulated by the Human Connectome Project data usage agreement.

This data includes samples of healthy adults in the age range of 22–35 years. There are 424 twins (172 males & 252 females) in this collection, 134 of which are same-sex identical twins (monozygotic: MZ) and 78 same-sex non-identical twins(dizygotic: DZ).

The MRI and fMRI images are captured with a customized 3T Siemens "Connectom" Skyra scanner having a 100 mT/m SC72 gradient insert and a standard Siemens 32-channel RF-receive head coil. The dataset contains at least one 3D T1w MPRAGE image and one 3D T2w SPACE image which has been acquired at 0.7 mm isotropic resolution. A multi-band EPI sequence with parameters of TR = 720 ms, 2 mm isotropic voxels, and multiband acceleration factor of 8 accuires whole-brain resting-state fMRI and task fMRI. An accurate cross-modal registration of structural and functional images in each subject is provided by accuiring spin echo field maps during both structural and fMRI scanning sessions [34–36].

The experimental task was an n-back working memory task in which participants were shown pictures of faces, places, objects, and body parts (Fig 1). In this experiment, we have two working memory modes: 2-back and 0-back, and each person participates in two task runs including 8 task blocks and 4 fixation blocks. Each task block contains 10 trials. In

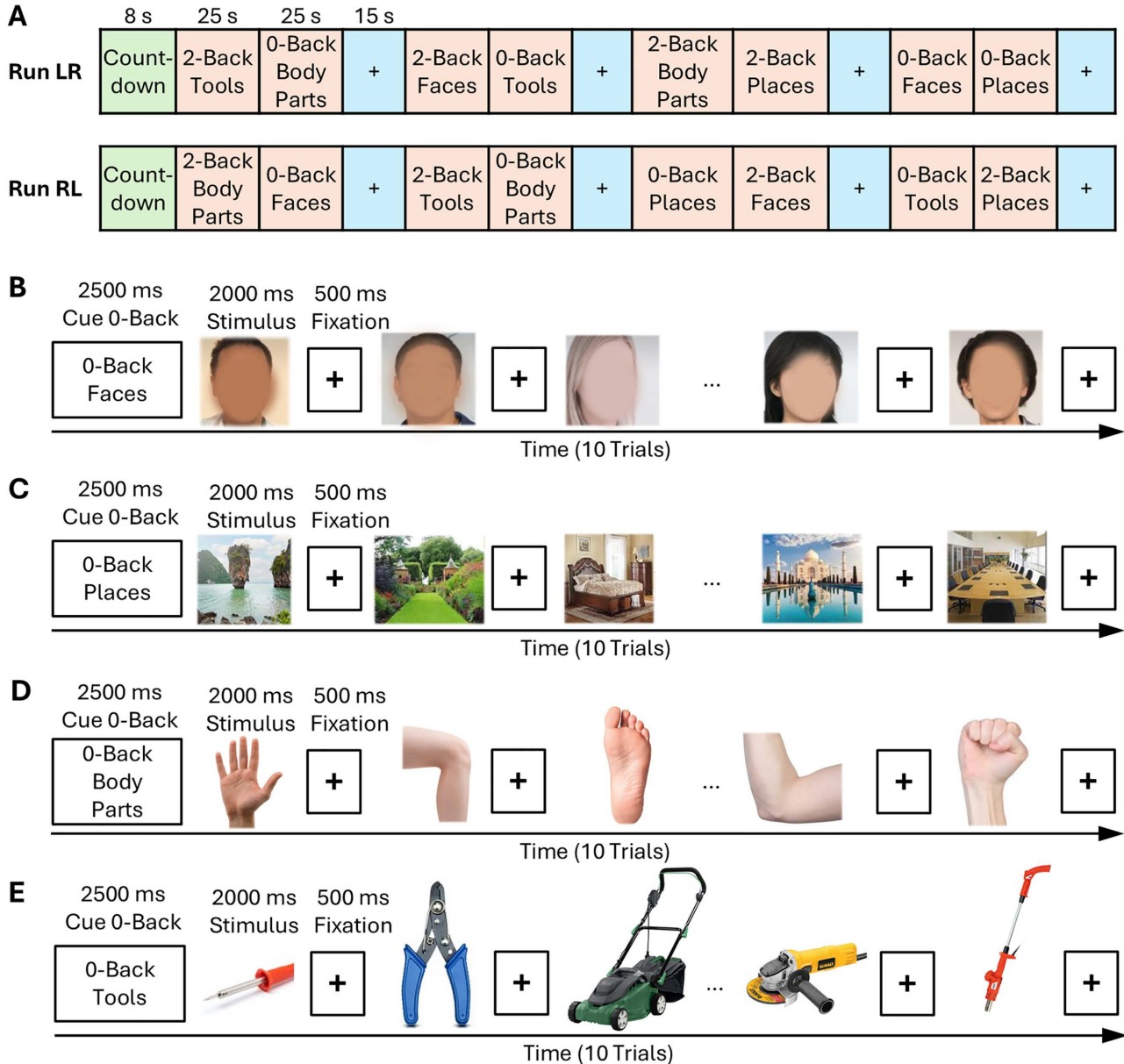

**Fig 1. The 2-back condition of the Human Connectome Project's Working Memory task [37].** utilized a block design with four different stimulus categories: faces, places, tools, and body parts. During the inter-trial intervals, a fixation cross (+) was presented. This figure provides examples of the broad range of stimuli used. At the start of each block, participants were shown a target cue and instructed to respond to any appearance of that stimulus during the block. Data were collected over two runs, with each run containing eight task blocks—four 0-back and four 2-back blocks. Each stimulus category (faces, places, etc.) had 20 trials for both 0-back and 2-back conditions. The task design involved a 2.5-second cue period followed by 10 trials where each stimulus was displayed for 2 seconds, interspersed with 0.5-second fixation periods, for a total of 25 seconds per run. Stimuli categories were shown in separate runs (faces, places, body parts, or tools), and in 50% of the runs, 2-back conditions were preceded by a 15-second fixation cross screen. Panels B–E depict examples of these 2-back runs.

each trial, the stimulus is displayed for 2 seconds, and then there is a 500-millisecond interval between trials. Among the stimuli, there are 2 target states and 2–3 non-target states. Also, the type of task is indicated by a 2.5 second guide at the beginning of the trial [37].

## Image processing

Preprocessing and analysis of the data is conducted by software tools published by the HPC [34]. Among them, the FSL packages are used for the preprocessing, Connectome Workbench command line tools and FreeSurfer packages [38] are used for the analysis of the data and the graphical user interface "wb_view" is used for the visualization of maps and generating ROIs (http://www.humanconnectome.org/software/connectome-workbench.html).

## Preprocessing of MRI images

preprocessing steps of MRI images include intensity normalization, brain extraction, bias field correction, co-registration of T1 and T2 images, tissue segmentation and spatial normalization. First, we should ensure that intensity values across scans and subjects are comparable by normalizing the intensity of brain tissue in the intensity normalization step. Next, in the brain extraction step, we should remove non-brain tissues (e.g., skull, scalp) from the image to focus on brain structures. Then we should correct intensity inhomogeneities caused by the MRI scanner, which can result in variability in intensity values across the image in the bias field correction. In the next step, in the co-registration of T1 and T2 images, we align T2w images to T1w images, often using affine or non-linear transformations to account for differences in position or head movement during scanning. Then, we separate different tissue types such as gray matter (GM), white matter (WM), and CSF. T1w images are typically used for this, and combined T1w/T2w can improve accuracy in differentiating between tissues. In the spatial normalization step, we Warp individual images to a standard anatomical space (e.g., MNI space) so that data across subjects can be compared.

## Analysis of structural data

Structural images T1w and T2w are analyzed to extract subcortical structures and to reconstruct cortical surfaces for each subject. After preprocessing step, we should do surface reconstruction. The program models the gray-white matter boundary, reconstructs the pial surface (the boundary between gray matter and CSF), and creates surface models. The cortical thickness and curvature can then be calculated. In the next surface refinement step, we should use a combination of curvature and intensity information from T1w images and FreeSurfer iteratively refines the cortical surfaces. The ratio of T1w/T2w images is often used to enhance tissue contrast and improve segmentation accuracy. This method is particularly effective in mapping myelin content in white matter. In tools like FreeSurfer (v6.0 and later), combining T1w and T2w images helps refine the pial surface and reduce errors in segmentation.The application of a multimodal surface matching (MSM) algorithm enabled us to accurately account for the cross-subject registration of cortical surfaces [39,40].

## Analysis of fMRI data

Several steps of preprocessing are performed on the fMRI images using the HCP pipeline [34]: First, the spatial distortions caused by the gradient nonlinearity and the inhomogeneity of the b0 field are corrected. Then a series of preprocessing is applied including field map-based unwarping of EPI images, motion correction, brain-boundary-based registration of EPI to structural T1w scan, nonlinear registration into MNI space, and grand-mean intensity normalization. Afterwards, the data are smoothed in the gray ordinates by a Gaussian Kernel with a 2 mm FWHM, and finally, an ICA+FIX is employed to up the structured noise and other artifacts.

Applying a temporal filter (Gaussian-weighted linear highpass filter with a cutoff of 200 s) to the fMRI data helps to suppress the low-frequency fluctuations and drifts in the data. The time series is prewhitend for removing temporal correlations.

## Genetic modeling

Genetic modeling of twin data using a univariate structural equation model is performed with the aid of statistical package OpenMx [41,42]. The ACE (Additive genetic, Common environment, and Unique environment) and ADE (Additive genetic,

Dominance genetic, and Unique environment) models are statistical approaches used in behavioral genetics to estimate the contribution of different factors to individual differences in traits. These models are particularly applied in twin studies to differentiate the influence of genetics and environment on traits.

The ACE model decomposes the variance of a trait into three components:1) Additive Genetic Factors (A) that represents the effect of individual genes summed across loci. These are the genetic influences that add up linearly and contribute to familial resemblance. 2) Common or Shared Environmental Factors (C) that represents environmental influences that make individuals growing up in the same family more similar to each other. It includes factors like socioeconomic status, parenting style, and family culture.3) Unique or Non-shared Environmental Factors (E) that represents environmental influences that contribute to differences between individuals, even those from the same family. It includes experiences unique to the individual, such as different peer groups or individual life events and also includes measurement error. For a given trait, the variance can be expressed as: $V = A + C + E$ Where A, C, and E are the variance components attributed to additive genetics, common environment, and unique environment, respectively.

The ADE model also decomposes variance into three components, but it differs from the ACE model by considering dominance genetic effects instead of shared environmental effects: 1) Additive Genetic Factors (A),2) Dominance Genetic Factors (D) that represents non-additive genetic effects, where the interaction between alleles at a locus can cause deviations from the additivity assumed in the ACE model. Dominance effects arise when the phenotype of a heterozygote is not intermediate between the phenotypes of the homozygotes.3) Unique or Non-shared Environmental Factors (E).

The variance can be expressed as: $V = = A + D + E$ Where A, D, and E represent the variance components due to additive genetics, dominance genetics, and unique environment, respectively.

Deciding whether to use an ACE or ADE model for analyzing twin data involves examining the patterns of trait correlation between monozygotic (MZ) and dizygotic (DZ) twins. So, we should calculate the intraclass correlations for the trait of interest in both MZ and DZ twins. $r_{MZ}$ represents the correlation of the trait between identical twins and $r_{DZ}$ represents the correlation of the trait between fraternal twins. Then we compare $r_{MZ}$ and $r_{DZ}$ for assess the relationship between correlations. If $r_{MZ} > 2 \times r_{DZ}$:This suggests non-additive genetic effects, such as dominance or epistasis. In this case, the ADE model might be more appropriate because it accounts for these dominance effects (D component). If $r_{MZ} \approx 2 \times r_{DZ}$: This indicates that additive genetic effects are likely dominant, and the shared environment (C) may have little influence. The ACE model might be more appropriate. If $r_{MZ} < 2 \times r_{DZ}$: This suggests that shared environmental factors (C) play a significant role in the trait variance. The ACE model should be considered, as it includes a parameter for shared environment. We calculate intraclass correlations for twin data and figured out that, ACE model is better describe our data than ADE model.

In the next step, the AE model is compared with the ACE model by comparing the calculated Akaike Information Criterion (AIC) metric of all voxels obtained from both fits [43].The lower AIC values of the fitted AE model in comparison to that of the fitted ACE model in the majority of voxels proves its advantage to model the measured data. We have therefore chosen the more appropriate AE model containing all genetic factors in A together with all environmental factors in E.

Based on a maximum likelihood estimation, the relative contribution of A and E factors is expressed by the regression path coefficients a and e. $a^2$ and $e^2$ parameters are the related variances and define by the mean square of $a^2$ and $e^2$ respectively and enabled the calculation of the percentage of variance explained by A in AE model as $a^2/(a^2 + e^2) \times 100$. The significance of genetic effects tested by omitting A from the model and using E submodel. The significance of the genetic effects is indicated by a substantial decrease in the goodness-of-fit for the $\chi^2$ statistics.

## Glasser parcellation for functional connectivity analysis

To define brain regions for functional connectivity analysis, we utilized the Glasser Human Connectome Project Multi-Modal Parcellation (HCP-MMP1.0) atlas [39]. This high-resolution parcellation scheme delineates 360 cortical regions (180 per hemisphere) based on a combination of functional, structural, and connectivity-based features, providing an anatomically and functionally informed framework for brain mapping.

Unlike traditional anatomical atlases such as the Automated Anatomical Labeling (AAL) atlas, which are based solely on macroanatomical landmarks, the Glasser parcellation integrates multiple neuroimaging modalities, including:

• Functional MRI (fMRI) (resting-state and task-based activation patterns)

• Myelin content mapping

• Cortical thickness

• Resting-state functional connectivity profiles

This multimodal approach ensures that each cortical region is defined not only by its structural features but also by its functional and connectivity properties, making it particularly well-suited for investigating task-based functional connectivity during cognitive paradigms such as the working memory task.

For our analysis, we extracted regional time series using the Glasser atlas, where each region corresponds to a distinct functional unit of the brain. The parcellated time series were subsequently used to compute both static and dynamic functional connectivity.

## Calculating correlation coefficients and normalization

After collecting time series data from multiple brain regions, we should ensure that the data from each brain region have a mean of zero and a standard deviation of one for normalization before calculating correlations. We used Pearson correlation coefficient method to measure the linear relationship between two time series from different brain regions. So we constructed a connectivity matrix where each element (i,j) represents the correlation coefficient between the time series of brain region i and brain region j.

For calculating dynamic connectivity, we applied a sliding window of fixed length to the time series data. The window slides over the entire time series, moving by a certain step size. Within each window, we calculated the correlation coefficient between the time series of different brain regions and construct a connectivity matrix resulting in a series of matrices that capture how connectivity changes over time. For Normalization Procedures, we Standardized each window's time series before calculating correlations, ensuring each segment has a mean of zero and a standard deviation of one.

## Calculating dynamic functional connectivity

We used windowing method for calculating dynamic connectivity. This method is a common approach to capturing time-varying changes in connectivity between brain regions over the course of an fMRI scan. It involves splitting the time series data from brain regions segments (or windows) and then calculating functional connectivity within each window. After preprocessing of time series data, a sliding window of fixed length is applied to the time series of each ROI. The window "slides" across the time series with a defined step size. The window length is crucial and typically depends on the specific dynamic phenomena being studied and the sampling rate of the fMRI data. Common window length is around 30 time points in fMRI studies. In the task-based studies, often use shorter windows to capture rapid changes in connectivity related to task performance. 30 TR in our study is equal to 21.6 seconds and we tried shorter window to assess the best result. Finally, we choose 18 seconds for window length and the step size of 3. For each window, the correlation between the time series of pairs of ROIs is computed using Pearson correlation method. This yields a connectivity matrix for each window, representing the functional connectivity between all pairs of ROIs during that time window. As the window slides across the time series, we compute the correlation matrices for each successive window. This results in a sequence of connectivity matrices, each reflecting the connectivity pattern within a particular time window. Once the dynamic functional connectivity matrices are obtained, techniques like k-means clustering can be used to group similar connectivity states into a smaller number of recurring "connectivity states" across time. The approaches include assessing the variance or mean of the dynamic connectivity matrices across windows is used to summarize dynamic behaviors.

## Dynamic state clustering and network analysis

Time-resolved functional connectivity matrices (subjects × windows × ROIs × ROIs) were averaged within Yeo 7-network modules to produce network × network matrices for each sliding window. Upper-triangular elements of these matrices were z-scored across all windows and subjects and clustered using k-means into two recurring.

connectivity states (State 1 and State 2). For each state, within-network connectivity was computed as the mean of connections among regions within the same network, and between-network connectivity as the mean of all inter-network connections. Temporal properties of each state were quantified by occupancy (proportion of windows spent in a state) and dwell time (mean duration of consecutive windows in a state). Group-level metrics were averaged across subjects, and paired t-tests were used to compare State 1 versus State 2.

## 3. Results

To increase the specificity of our analyses, we performed all genetic and connectivity assessments using the 2-back – 0-back differential phenotype. This contrast removes low-level perceptual and motor components shared across both tasks and isolates neural processes uniquely related to working-memory load. By reducing inter-individual variability and eliminating task-general effects, the differential approach provides a cleaner and more interpretable phenotype for twin modeling, improving sensitivity to detect heritable influences on static and dynamic connectivity.

## Static functional connectivity

Static functional connectivity measures the time-averaged correlation between brain regions, assuming stable interactions throughout the task. In contrast, dynamic functional connectivity captures fluctuations over time, allowing us to examine how connectivity patterns evolve in response to cognitive demands.

First, we compared static connectivity across the entire brain in MZ and DZ twins. The BOLD time series was extracted from each of the 360 Glasser ROIs, and partial correlations were computed between regions to estimate functional relationships. After applying multiple comparison correction (Bonferroni correction, p-value = 0.03), 58% of the connections showed significant differences between MZ and DZ twins..

To quantify genetic influence, we applied the ACE model, which estimates the proportion of variance explained by genetic (A), shared environmental (C), and unique environmental (E) factors. Model selection using Akaike's Information Criterion (AIC) indicated that the AE model provided a better fit than the ACE model for most connections, meaning that shared environmental effects (C) were negligible.

To achieve accurate results, we selected working memory-related regions in the Glasser parcellation and performed genetic analysis on these areas. The Human Connectome Project (HCP) study on the 2-back working memory task (Barch et al., 2013) identified significant activation in key brain regions associated with higher cognitive processing and working memory functions. These regions.

primarily belong to the Frontoparietal Control Network (FPCN) and Dorsal Attention Network (DAN), which are crucial for cognitive control and attentional processing.

The Prefrontal Cortex (PFC) plays a central role in working memory, executive function, and cognitive control. In the Glasser parcellation, key working memory-related areas include Dorsolateral Prefrontal Cortex (DLPFC) – Area 46, Lateral Prefrontal Cortex (LPFC) – Area 9, Frontal Eye Fields (FEF) – Area 8 and Anterior Prefrontal Cortex (Frontopolar Cortex) – Area 10.

The Parietal Cortex supports working memory storage, attention, and visuospatial working memory. Relevant Glasser regions include Intraparietal Sulcus (IPS) & Superior Parietal Lobule (SPL): LIPd, LIPv (*Lateral Intraparietal Area*), AIP (*Anterior Intraparietal Area*), MIP (*Medial Intraparietal Area*), PGp, PGi (*Posterior Parietal Cortex*) and Precuneus: POS1, POS2, 7Am, 7Pm.

The Cingulate Cortex is also involved in working memory, particularly in cognitive control and attention regulation. Key working memory-related regions in the Glasser parcellation include Dorsal Anterior Cingulate Cortex (dACC): a24pr, p24pr, d32 and Posterior Cingulate Cortex (PCC): d23ab, v23ab, 31a, 31pd.

Additionally, the Temporal and Insular Cortex are implicated in memory processing and multimodal integration. Important regions include Middle Temporal Gyrus & Superior Temporal Gyrus: PHT, TE1p, TE1m, TE2a and Insular Cortex: Ig, AVI, FOP.

Analysis of the AE twin model on these working memory related regions revealed that genetic factors contributed moderately—but in a highly selective manner—to static (time-averaged) functional connectivity (Table 1). Although the majority of cortical edges showed low heritability, a specific subset of connections demonstrated moderate genetic effects (A = 0.298–0.328), largely confined to heteromodal association cortex. The most heritable edge was identified between TE1m and PGi (A = 0.328), two key nodes of the Default Mode Network (DMN), emphasizing a notable genetic contribution to intrinsic DMN coupling.

Other highly heritable edges included links between frontal–cingulate control regions (8Ad–p24pr), temporal–frontal association pathways (p9-46v–9m; 46–TE2a; p9-46v–TE2a), and posterior DMN subsystems (31pd–PHT; 31a–v23ab). Collectively, these results indicate that genetic factors most strongly influence static connectivity within DMN, frontocingulate control networks, and temporal association cortex, whereas primary sensory–motor systems showed minimal heritability. The average heritability across the top edges (~31.6%) suggests that static FC reflects a partially heritable, trait-like component of functional architecture.

## Heritability of mean dynamic connectivity (dFC–Mean)

Genetic effects on mean dynamic connectivity were moderate (A ≈ 0.23–0.30) and exhibited a distinct spatial profile compared to static FC (Fig 2). The strongest genetic influence was obseed for the connection LIPd–31a (A = 0.303), linking a dorsal attention region with a posterior DMN hub (Table 2). This indicates that inherited factors help regulate the baseline dynamic balance between attention-related and internally oriented networks, a core mechanism underlying working-memory control.

Other top heritable edges mainly involved interactions among the DMN, frontoparietal control system, and anterior cingulate/salience network, including 10pp–PGp, a24pr–10v, PGi–d23ab, and 8Av–PGi. These patterns highlight genetic influences on cross-network coordination, particularly mechanisms that support transitions between internally and externally directed cognitive states.

Additionally, several heritable edges (e.g., a9-46v–9m, 31pd–a9-46v) reflected genetic contributions to frontal executive and midline DMN dynamics, supporting the idea that stable aspects of dynamic integration within executive systems are partly genetically determined.

**Table 1. Heritability results for static functional connectivity.**

| ROI 1 | ROI 2 | A-effect |
| --- | --- | --- |
| TE1m | PGi | 0.328 |
| 8Ad | p24pr | 0.322 |
| 8C | TE2a | 0.316 |
| 31pd | PHT | 0.31 |
| 31a | v23ab | 0.306 |
| p9-46v | 9m | 0.305 |
| 46 | TE2a | 0.304 |
| p9-46v | TE2a | 0.304 |
| TE2a | POS2 | 0.301 |
| TE2a | 10pp | 0.298 |

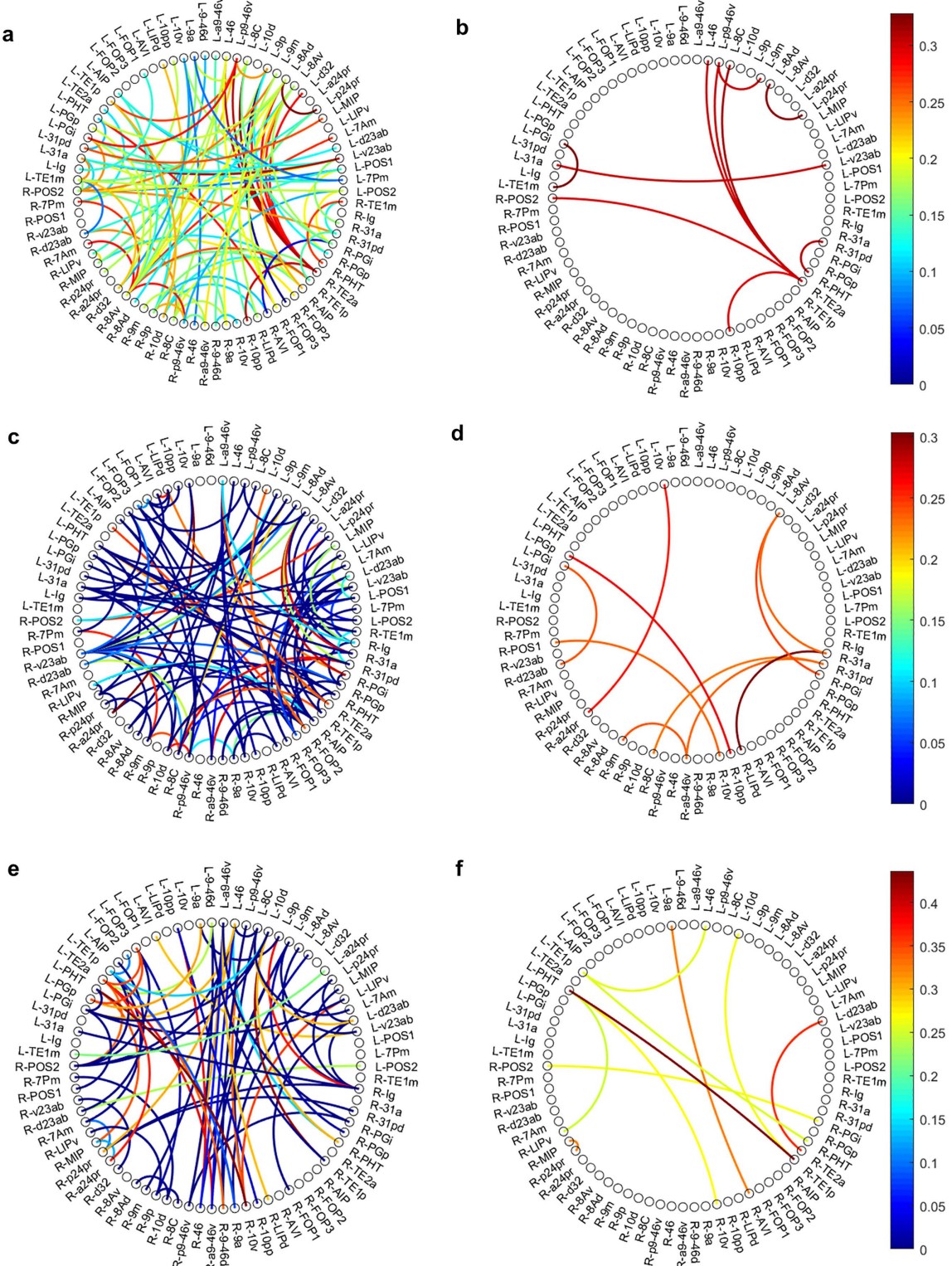

**Fig 2. Circulargraphs of genetic factor (A) of functional connectivity in working memory related regions of glasser parcellation. a)** Circulargraphs of A factor of static connectivity for working memory related areas **b)** The 10-top A factor of static connectivity for working memory related areas. **c)** Circulargraphs of A factor of mean of dynamic connectivity for working memory related areas. **d)** The 10-top A factor of mean of dynamic connectivity

for working memory related areas. **e)** Circulargraphs of A factor of variance of dynamic connectivity for working memory related areas. **f)** The 10-top A factor of variance of dynamic connectivity for working memory related areas. In each part, genetic factor (A) of functional connectivity is computed using AE modelling and after multiple comparisons testing (p-value = 0.05), it is displayed on the circular graph of Glasser parcellation ROIs.

**Table 2. Heritability results for mean of dynamic connectivity.**

| ROI1 | ROI2 | A_effect |
|------|------|----------|
| R-LIPd | R-31a | 0,303 |
| R-10pp | L-PGp | 0,269 |
| R-a24pr | L-10v | 0,259 |
| L-8Av | R-PGi | 0,243 |
| L-PGi | R-d23ab | 0,239 |
| R-a9-46v | R-9m | 0,237 |
| R-31a | L-8Av | 0,234 |
| R-10v | R-POS1 | 0,234 |
| R-31pd | R-a9-46v | 0,233 |
| R-31pd | R-8C | 0,230 |

## Heritability of dynamic connectivity variability (dFC–Variance)

The variance of dynamic connectivity displayed the strongest genetic effects among all three phenotypes, with the top edges showing substantial heritability (A = 0.25–0.44). The most genetically influenced connection was PHT-TE1p (A = 0.436), linking occipito-temporal multimodal regions involved in high-level perceptual integration (Table 3). This suggests that moment-to-moment neural flexibility in multisensory pathways is strongly genetically regulated.

According to Table 3, strong genetic effects were also observed for edges connecting temporal association cortex (TE2a) with posterior DMN (d23ab), and for connections within cingulo-opercular and frontocingulate control networks, such as MIP–p24pr and AVI–9a. These regions support sustained attention, salience detection, error monitoring, and uncertainty evaluation, indicating that inherited factors shape dynamic control processes essential for high-load working-memory tasks.

Additional genetically influenced edges involved interactions between temporal areas (TE1p) and prefrontal control regions (10d, 46), as well as dorsal attention nodes (POS2, LIPv), reinforcing the idea that temporal–prefrontal communication and attentional reorientation dynamics have a meaningful genetic basis.

**Table 3. Heritability results for variance of dynamic connectivity.**

| ROI1 | ROI2 | A_effect |
|------|------|----------|
| L-PHT | R-TE1p | 0,436 |
| R-TE2a | L-d23ab | 0,361 |
| R-MIP | R-p24pr | 0,327 |
| R-AVI | L-9a | 0,325 |
| L-PHT | R-10v | 0,272 |
| R-POS2 | R-PGi | 0,268 |
| L-10d | R-TE1p | 0,267 |
| L-46 | L-TE1p | 0,260 |
| R-PHT | L-TE1p | 0,257 |
| L-PHT | R-LIPv | 0,251 |

Across all three connectivity metrics, genetic effects consistently concentrated in heteromodal association cortex, particularly: Default Mode Network (DMN), Fronto-cingulate and frontoparietal control systems and Temporal multimodal association regions. These areas mediate self-referential processing, executive control, multisensory integration, and cognitive flexibility—functions known to exhibit substantial interindividual variation and heritable component.

To further examine which anatomical regions contribute most strongly to the genetically significant functional connections, we quantified the proportion of significant edges associated with each cortical region across the three connectivity measures. As illustrated in Fig 3, the distribution of genetically influenced connections varied systematically across regions and across connectivity metrics. Beyond the dominant contribution of the dorsolateral prefrontal cortex (DLPFC), lateral prefrontal cortex (LPFC), and frontal eye fields (FEF), key frontoparietal hubs involved in executive control, working memory manipulation, and top-down attentional guidance, several additional regions showed notable levels of genetic influence. For Static FC, the posterior cingulate cortex (PCC), a central node of the default-mode network implicated in internally directed cognition and integration of self-referential information, and the temporal cortex, which supports higher-order perceptual processing and memory-related functions, exhibited the highest proportions of genetically significant connections, each accounting for approximately 30% of all significant edges linked to their respective regions. For dFC-Mean, the intraparietal sulcus/superior parietal lobule (IPS/SPL) regions, core components of spatial attention and visuomotor integration, along with the PCC, similarly demonstrated elevated contributions, again reaching about 30%. In

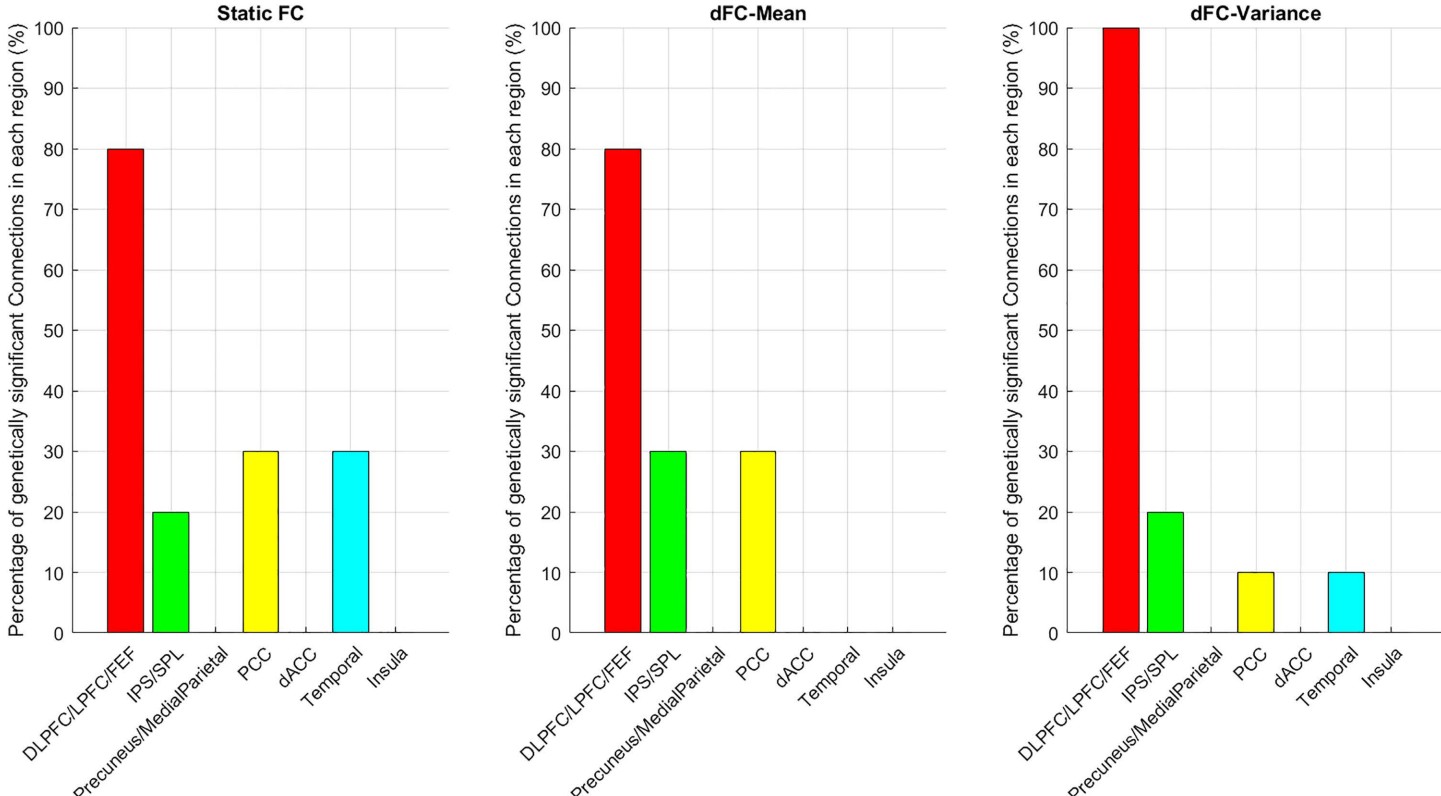

**Fig 3. Percentage of genetically significant connections involving each anatomical brain region for three functional connectivity measures.** The figure shows three subplots corresponding to **(A)** Static FC, **(B)** dynamic FC mean (dFC-Mean), and **(C)** dynamic FC variance (dFC-Variance). Each bar represents the proportion of significant connections in which at least one end of the connection belongs to the specified anatomical region. Colors indicate different anatomical regions: DLPFC/LPFC/FEF, IPS/SPL, Precuneus/Medial Parietal, PCC, dACC, Temporal, and Insula. This visualization highlights the contribution of each anatomical region to the overall pattern of genetically significant connections across the three connectivity measures.

contrast, for dFC-Variance, the IPS/SPL regions showed the largest contribution, comprising roughly 20% of genetically significant connections. Collectively, these findings underscore that genetic effects are not randomly distributed across the cortex but preferentially target specific hubs whose functional roles align with high-level cognitive control, attentional regulation, and the integration of internally and externally driven information.

As summarized in Table 4, the three phenotypes examined in this study exhibited distinct but partially overlapping heritability profiles, reflecting different dimensions of genetically shaped functional brain organization. Static functional connectivity showed moderate heritability (A≈0.30), with its spatial signature concentrated in the default mode network (DMN) and frontal–temporal pathways. This pattern suggests that the stable, trait-like architecture of intrinsic brain networks is substantially shaped by genetic factors. The mean of dynamic connectivity (dFC–Mean) also demonstrated moderate heritability (A≈0.23–0.30), primarily involving interactions between the DMN, attention networks, and frontal control systems. This indicates that genes contribute not only to static baseline organization but also to the more stable components of network coordination that support cognitive engagement. In contrast, the variance of dynamic connectivity (dFC–Variance) showed the strongest heritability (A≈0.25–0.44), with contributions spanning temporal–DMN coupling, cingulo-opercular systems, executive networks, and multisensory pathways. This heightened genetic influence suggests that moment-to-moment neural flexibility, an essential feature of dynamic adaptation during cognitive processing, is particularly sensitive to genetic variability.

### Stable dynamic brain states: Network-level and genetic differences

To characterize large-scale dynamic functional organization during the working memory task, we analyzed time-resolved ROI connectivity matrices (subjects × windows × ROIs × ROIs) using a two-state k-means clustering framework. For each window, connectivity matrices were extracted, averaged within Yeo-based functional networks (Visual, Somatomotor, Dorsal Attention, Ventral Attention/Salience, Limbic, Frontoparietal Control, DMN), and assigned to one of two recurring dynamic states (State 1 and State 2). For each subject, we computed (i) mean within- and between-network connectivity for each state, (ii) occupancy rate (proportion of windows spent in each state), and (iii) dwell time (mean consecutive windows remaining in a given state). Paired t-tests were used to compare State 1 vs. State 2 across subjects.

### Within-network connectivity differences between dynamic states

Group-level analyses of within-network connectivity revealed that most networks exhibited similar organization across the two dynamic states, but selective differences emerged in specific systems (Fig 4). The Visual network showed a significant difference (p=0.043), and the Limbic network exhibited a trend-level effect (p=0.05), with both networks demonstrating stronger internal coherence in State 1 compared with State 2. According to Fig 4, no significant differences were observed in the Somatomotor, Dorsal Attention, Ventral Attention/Salience, Frontoparietal Control, or Default Mode networks (p>0.26). Notably, most networks displayed negative mean values, indicating reduced within-network coherence in State 2 relative to State 1, while the DMN was the only network showing a relative increase. Overall, these findings

**Table 4. Differences between static, dynamic-mean, and dynamic-variance heritability.**

| Phenotype | Heritability Strength | Spatial Profile | Interpretation |
|---|---|---|---|
| **Static FC** | Moderate (A≈0.30) | DMN+frontal/temporal pathways | Reflects stable, trait-like baseline architecture influenced by genetic factors |
| **dFC–Mean** | Moderate but slightly lower (A≈0.23–0.30) | DMN–attention+DMN–frontal control interactions | Genes contribute to stable aspects of network coordination during cognitive engagement |
| **dFC–Variance** | **Strongest (A≈0.25–0.44)** | Temporal–DMN, cingulo-opercular, executive, and multisensory networks | Suggests neural flexibility and moment-to-moment adaptability are highly heritable |

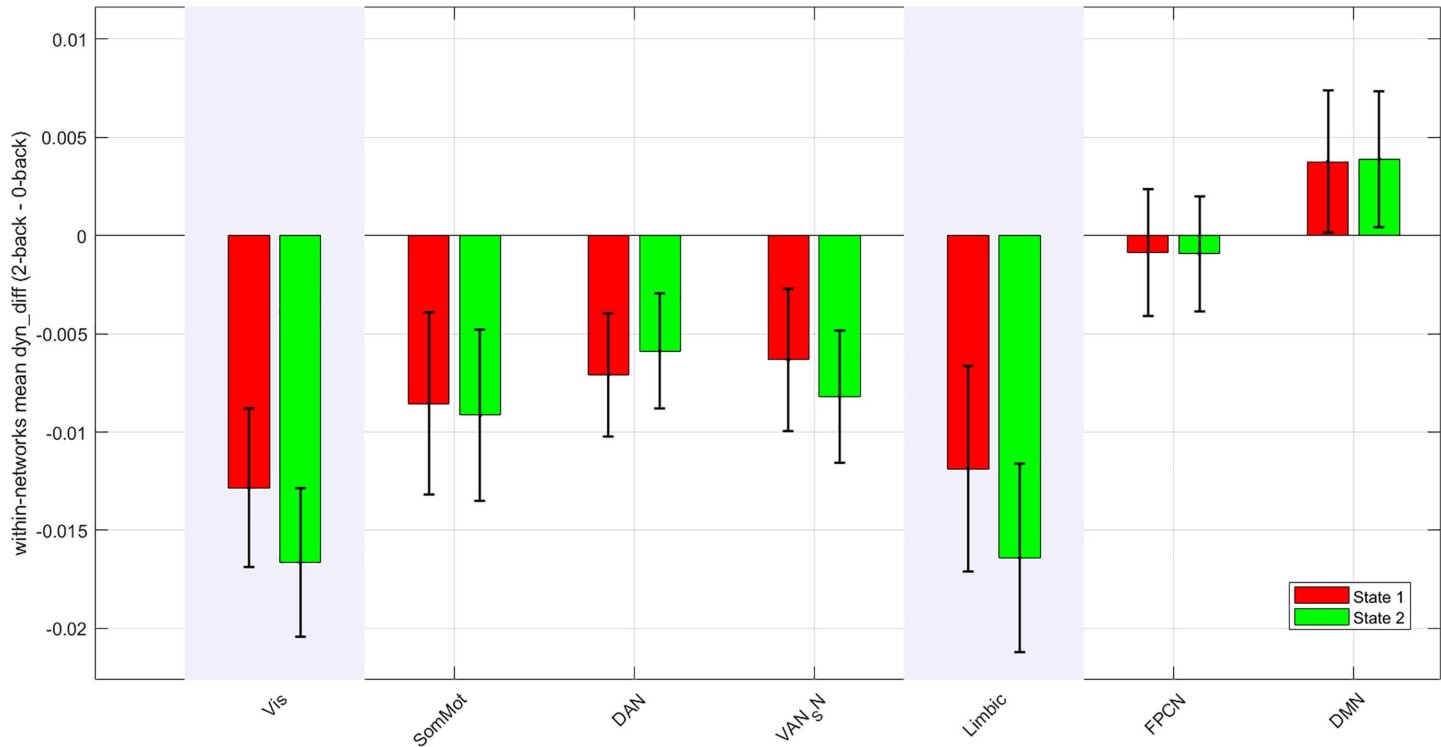

**Fig 4. Within-network connectivity differences across dynamic states.** Bar plots show mean within-network connectivity (mean ± SEM) for each large-scale functional network in State 1 and State 2. Among all networks, only the Visual and Limbic networks exhibited significant differences between states(p-value = 0.05), with both showing stronger internal coherence in State 1. All other networks—including Somatomotor, Dorsal Attention, Ventral Attention/Salience, Frontoparietal Control, and Default Mode—showed no significant state-dependent differences.

suggest that the primary distinction between the two dynamic states lies in the segregation of sensory–perceptual (Visual) and affective–associative (Limbic) systems, whereas other large-scale networks remain stable across dynamic modes.

## Between-network connectivity: stability across states

Between-network interactions remained statistically stable across states. None of the tested network pairs, including FPCN–DAN, FPCN–DMN, and VAN/SN–FPCN, showed significant differences ($p > 0.44$). The absence of between-network changes suggests that the principal distinction between the two dynamic modes lies in within-network organization rather than large-scale reconfiguration between systems.

## Temporal characteristics of dynamic states

Temporal metrics further distinguished the two states. State 2 was markedly dominant, accounting for 75.6% of all windows, whereas State 1 occupied only 24.4% of the time. Dwell time analyses revealed that State 2 was also substantially more stable, with a mean dwell of 3.98 consecutive windows, compared with only 1.29 windows for State 1.

Taken together, these temporal profiles indicate that State 2 represents the default and more persistent dynamic configuration during task processing, while State 1 appears as a transient, short-lived, more segregated mode that intermittently emerges.

To quantify heritability, monozygotic (MZ) and dizygotic (DZ) twin pairs were compared on summary metrics derived from each connectivity state. No dynamic connectivity metric showed a positive genetic pattern (rMZ > rDZ). Heritability

estimates were near zero or negative, indicating little to no genetic influence. These findings suggest that the state structure, state switching behavior, and within-network coupling during working-memory dynamics are largely shaped by environmental or task-driven variability, not inherited factors.

## 4. Discussion

In this study, we explored the heritability of static and dynamic functional connectivity during a 2-back and 0-back working memory task in twin pairs (MZ vs. DZ), providing insights into how genetic and environmental factors contribute to neural network communication under cognitive load. By focusing on task-based connectivity — rather than solely resting state — we extend the literature into the functional domain of working memory and cognitive control.

Our analyses reveal a clear gradient of heritability across three connectivity phenotypes: static functional connectivity (FC), mean dynamic connectivity (dFC–Mean) and variance of dynamic connectivity (dFC–Variance). Using an AE twin model, we found moderate heritability (A ≈ 0.30) confined to a selective subset of heteromodal association regions, notably within the Default Mode Network (DMN), fronto-cingulate control systems, and temporal association cortex. Primary sensory-motor systems demonstrated negligible genetic influence. This pattern is consistent with previous findings in resting-state connectivity (heritability ≈ 15–42%) [10,16,17]. For dFC-Mean, genetic effects were moderate (A ≈ 0.23–0.30) with a distinct spatial topology compared to static FC. The most heritable edge (R-LIPd–R-31a, A ≈ 0.303) linked a dorsal attention network node with a posterior DMN hub. Additional high-heritability edges involved DMN–frontal control and attention–control interactions (e.g., R-10pp–L-PGp, R-a24pr–L-10v). These results suggest that baseline dynamic coordination between attention/executive systems and internal mentation networks is partially under genetic control. Finally, we observed the highest heritability in dFC-variance (A ≈ 0.25–0.44). The most pronounced genetic influence was seen in edges linking temporo-parietal association regions (L-PHT–R-TE1p, A ≈ 0.436), fronto-parietal/cingulo-opercular control networks (R-MIP–R-p24pr; R-AVI–L-9a), and dorsal attention nodes (POS2, LIPv). These findings indicate that neural flexibility – the capacity of brain networks to reconfigure over time – is strongly driven by genetic architecture.

These results are consistent with findings from resting-state dFC studies, where greater heritability has been observed for time-varying patterns compared to static connectivity [23]. The ability of brain networks to rapidly reconfigure in response to cognitive demands may be under genetic control, reflecting inherited differences in neurotransmitter systems, synaptic plasticity, and white matter microstructure [44].

To contextualize the strength of the observed heritability effects, it is important to situate these findings within the broader neurogenetic literature. Structural brain phenotypes such as cortical surface area and thickness typically exhibit high heritability (40–80%), whereas functional traits-including task-evoked activation and resting-state connectivity-generally fall within a lower range (10–30%). Within this framework, the ~30% heritability observed for the mean of dFC represents the upper bound of what is commonly reported for functional measures, while the high heritability of dFC variance places it closer to structural traits traditionally considered strong endophenotypic candidates. Thus, our interpretation of dFC variance as a promising endophenotype is grounded not in subjective terminology but in comparative evidence across established neurobiological phenotypes.

The within-network analysis revealed a consistent pattern in which most large-scale functional networks exhibited reduced internal coherence across dynamic states, whereas only the Default Mode Network (DMN) showed a relative increase in within-network coupling. This profile aligns with well-established task-based fMRI results showing that cognitive load typically suppresses intrinsic connectivity within sensory, attentional, and control networks while simultaneously increasing their cross-network integration. In contrast, the DMN becomes more internally coherent during high-demand conditions, reflecting its characteristic suppression and functional segregation from task-positive systems. The negative values observed in most networks therefore likely reflect task-induced decreases in network segregation, whereas the positive shift in the DMN indicates enhanced internal stability during task engagement. This divergence between DMN and non-DMN systems fits with prior reports demonstrating that task-evoked dynamics preferentially reorganize the

communication structure of attention and control networks while promoting a more segregated configuration of the DMN. Overall, our within-network results support the interpretation that the two dynamic states differ primarily in their levels of network segregation, with the DMN uniquely expressing increased coherence relative to other networks.

Working memory is a core cognitive function that varies significantly between individuals, and our results suggest that some of this variability may stem from genetically influenced differences in neural connectivity patterns. The fact that dFC variability shows the highest heritability suggests that the flexibility of functional interactions between brain regions may be a genetically mediated trait linked to cognitive adaptability. This supports theories proposing that efficient cognitive functioning depends on the brain's ability to dynamically shift between different functional states [45].

Additionally, these findings may provide insight into the neural efficiency hypothesis, which posits that greater cognitive efficiency is associated with more flexible and dynamic brain networks rather than rigidly fixed connections [46]. Individuals with highly heritable, flexible connectivity patterns might demonstrate superior working memory performance, though further research is needed to directly link dFC variability to behavioral outcomes.

The role of dFC variability in neuropsychiatric and neurodevelopmental disorders is an emerging area of research. Abnormal fluctuations in dynamic connectivity have been reported in schizophrenia, ADHD, autism spectrum disorder, and depression [47,48]. Notably, increased variance in dFC has been observed in schizophrenia, suggesting that abnormal network flexibility may underlie cognitive deficits in psychiatric conditions.

Our findings suggest that genetic factors may contribute to these alterations in dFC variability, which could help identify potential genetic risk markers for cognitive dysfunction. If dFC variance is indeed an endophenotype, future studies could explore its role in early identification of at-risk individuals or personalized interventions targeting neural network flexibility.

Moreover, understanding the heritability of dynamic connectivity could inform research on cognitive aging and neurodegenerative diseases. Some studies suggest that aging is associated with reduced dFC variability, potentially linked to cognitive decline [49]. If dFC variability is partially genetically determined, individuals with a strong genetic predisposition for high dFC variability may be more resilient to age-related cognitive decline.

Given that dFC variability appears to be a heritable trait, it could be explored as a biomarker for cognitive vulnerability in clinical populations. Future research could investigate identifying individuals with atypical heritable dFC patterns (e.g., children with a family history of ADHD or schizophrenia) could allow for earlier interventions before symptoms fully emerge. Understanding genetic influences on dFC may help predict which individuals respond best to cognitive training, medication, or neuromodulation therapies. Since dFC reflects neural flexibility, interventions targeting real-time brain connectivity adjustments (e.g., neurofeedback, cognitive training) may be particularly beneficial for individuals with genetically determined deficits in network flexibility.

To further explore the genetic basis of working memory deficits, future studies should investigate heritability of dFC in clinical populations with neurodevelopmental or psychiatric conditions and examine longitudinal heritability patterns, assessing whether dFC variability remains stable across development or changes with disease progression. Future studies should explore gene-brain-behavior relationships, linking genetic variants to working memory performance and dFC alterations and use multimodal neuroimaging (fMRI + EEG + DTI) to integrate functional and structural genetic influences on working memory.

A core advance of our analyses (including a differential phenotype: 2-back minus 0-back) is the tighter alignment of our heritability results with the Triple-Network Model (Salience Network – Frontoparietal Network – Default Mode Network) [50–59]. According to this model, efficient cognitive control and higher-order cognition depend on dynamic interactions and switching among these three networks. We demonstrate that the highest heritability resides in edges connecting SN–FPN and FPN–DMN regions, consistent with the hypothesis that the SN acts as a central "switch" hub initiating network transitions in response to task-demand [50,52,53]. Our findings suggest that individual differences in the flexibility and efficiency of these network transitions are genetically mediated. Although *static FC* reflects a trait-like connectivity backbone, with moderate genetic influence but little indication of SN-mediated switching, *dFC–Mean* captures average dynamic interactions

                                                                 

but does not index temporal variability or switching per se. *dFC–Variance* best corresponds to the Triple-Network Model in a genetic sense, as it represents network reconfiguration and switching phenomena central to SN–FPN–DMN interactions. For functional and Cognitive Relevance, genetically influenced SN–FPN variance edges may relate to how efficiently individuals detect and respond to salient cognitive demands (e.g., WM load switches) [51,54]. Genetically influenced FPN–DMN variance edges may underpin effective integration between executive control and self-referential systems, supporting adaptability in working memory and strategic cognition [55–57].Importantly, our twin results suggest that network flexibility — rather than static connectivity — may serve as a neurobiological endophenotype for cognitive performance and risk resilience in psychopathology. Given the established involvement of SN–FPN–DMN dysregulation in psychiatric disorders (ADHD, schizophrenia, depression) [52,58,59], our finding that variance of these interactions is highly heritable raises promising translational implications for identifying genetically mediated network vulnerabilities.

This study extends prior resting-state heritability research by focusing on a *task-based working memory paradigm*, thereby assessing functional connectivity under cognitive demand. Despite the advances, several limitations should be acknowledged.Our sample consisted exclusively of twin pairs (MZ & DZ), which may limit the generalizability to the broader population of unrelated individuals. Future studies should incorporate large, population-based cohorts and polygenic risk measures.The Glasser atlas, while high resolution, remains a structural parcellation; future work could leverage subject-specific or multimodal atlases to better map SN/FPN/DMN nodes. Our phenotype derived from a single working memory task (2-back minus 0-back); studies using event-related designs manipulating salience explicitly would enable more direct tests of SN switching. Longitudinal and developmental designs are needed to assess how genetic influences on network flexibility evolve over the lifespan or with neurodegenerative processes.our analyses isolate the working-memory–specific component of connectivity by contrasting 2-back and 0-back conditions, the current design does not disentangle potential modality-specific contributions (e.g., face, place, or tool stimuli). Future work could extend this framework by examining differential heritability across stimulus categories, allowing assessment of whether the high heritability of dFC variance reflects a general mechanism of executive load or whether certain perceptual modalities contribute more strongly to genetic modulation. Such multimodal analyses may reveal modality-dependent genetic signatures that complement the load-driven effects observed here.

In summary, our results support the hypothesis that genetic factors shape not only the static architecture of brain networks but — more powerfully — the temporal architecture of network dynamics, particularly within the SN–FPN–DMN axis. We propose that dynamic variability of connectivity may serve as a more sensitive, theory-driven endophenotype for cognitive control and higher-order function, with broad implications for understanding individual differences and psychiatric risk.

## 5. Conclusion

Our findings demonstrate that genetic factors exert moderate influence on static and mean dynamic functional connectivity, but most strongly shape the variance of dynamic connectivity, reflecting neural flexibility and network reconfiguration. This highlights dynamic variability—particularly within SN–FPN–DMN interactions—as a heritable neural phenotype that may underlie individual differences in cognitive control, working memory, and susceptibility to neuropsychiatric conditions. Overall, dynamic connectivity variance emerges as a sensitive, theory-driven endophenotype for future investigations into the genetic basis of cognition and brain network adaptability.

Our findings suggest that future research should focus on exploring the role of dynamic connectivity in other tasks and clinical conditions, as well as advanced techniques like Psychophysiological Interaction (PPI) analysis to further refine our understanding of task-specific brain network dynamics.

## Author contributions

**Conceptualization:** Mohammad Reza Daliri.

**Methodology:** Maryam Fatemi, Mohammad Reza Daliri.

**Resources:** Mohammad Reza Daliri.

**Software:** Maryam Fatemi.

**Supervision:** Mohammad Reza Daliri.

**Writing – original draft:** Maryam Fatemi.

**Writing – review & editing:** Maryam Fatemi, Mohammad Reza Daliri.

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
