## [Decision Letter · Decision Letter 0]

8 Oct 2025

Dear Dr. Daliri,

Thank you for submitting your manuscript to PLOS ONE. After careful consideration, we feel that it has merit but does not fully meet PLOS ONE’s publication criteria as it currently stands. Therefore, we invite you to submit a revised version of the manuscript that addresses the points raised during the review process.

We look forward to receiving your revised manuscript.

Kind regards,

Kenji Tanigaki, Ph.D., M.D.

Academic Editor

PLOS ONE

Additional Editor Comments (if provided):

Reviewers' comments:

Reviewer's Responses to Questions

**Comments to the Author**

1. Is the manuscript technically sound, and do the data support the conclusions?

Reviewer #1: Yes

Reviewer #2: Partly

2. Has the statistical analysis been performed appropriately and rigorously?

Reviewer #1: Yes

Reviewer #2: I Don't Know

3. Have the authors made all data underlying the findings in their manuscript fully available?

Reviewer #1: Yes

Reviewer #2: No

4. Is the manuscript presented in an intelligible fashion and written in standard English?

Reviewer #1: Yes

Reviewer #2: No

Reviewer #1: This is a nice paper where the authors applied genetic modeling to both static and dynamic connectivity measures and show low heritability (16%) for static connectivity within the Frontoparietal Control Network (FPCN) and Dorsal Attention Network (DAN), but moderate heritability (30%) in the mean dynamic connectivity between the Temporal Gyrus and Frontal Eye Fields, Anterior PFC and Frontal Eye Fields, Lateral PFC and Lateral Intraparietal Area, and Temporal Gyrus and Lateral PFC. They further found that the variance of dynamic connectivity exhibits the highest heritability (78%), particularly in connections between the Dorsal Anterior Cingulate Cortex and Lateral PFC, as well as the Lateral PFC and Posterior Cingulate Cortex.

The authors conclude that the findings highlight that dynamic connectivity variability during working memory tasks is more

strongly influenced by genetic factors than static connectivity.

Overall, the paper is clearly written, however, the Discussion section seems weak. The authors talk about the SN, DMN, and FPN, and also show that the variance of the dynamic connectivity between SN-FPN and FPN-DMN is very strongly influenced by genetic factors, but do not discuss these in light of the triple network model recently proposed, which posits that dynamic interactions and switching (i.e., variance in authors' current manuscript) between the SN, DMN, and FPN is foundational for many higher order processing in the human brain. The authors should add a separate sub-section in the Discussion section to discuss their findings in the context of the triple network model using the following references:

[1] https://www.nature.com/articles/s41467-021-23509-x

[2] https://academic.oup.com/cercor/article/26/5/2140/1754229

[3] https://elifesciences.org/articles/99018

[4] https://academic.oup.com/cercor/article/30/10/5309/5840454

[5] https://www.sciencedirect.com/science/article/pii/S1053811922000568

[6] https://academic.oup.com/cercor/article/32/23/5343/6524032

[7] https://www.sciencedirect.com/science/article/pii/S0010945221003701

[8] https://academic.oup.com/cercor/article/34/7/bhae287/7718277

[9] https://link.springer.com/article/10.1007/s00429-010-0262-0

[10] https://elifesciences.org/articles/76702

The authors should also discuss whether the static or the dynamic connectivity is better aligned with the triple network model in the context of genetic influences. This will put the authors' findings in a stronger theoretical grounding.

Reviewer #2: The present study utilized the HCP1200 dataset to investigate the heritability of both static and dynamic functional connectivity (dFC) in twins during an N-back working memory task. The findings indicate low heritability (16%) for static functional connectivity, but a moderate genetic influence (30%) on the mean of dynamic functional connectivity (dFC), and an even higher heritability (78%) for the variance of dFC. This suggests that fluctuations in brain connectivity over time are more genetically driven than static connectivity measures, supporting the notion that network flexibility may serve as a potential endophenotype for cognitive traits.

While the study addresses a potential gap in the literature, several major conceptual, methodological, and presentational concerns limit the impact and interpretability of its findings.

The justification for focusing on the working memory task was limited to the statement that "heritability estimates in task-based functional connectivity remain underexplored." This claim seems contradictory to the following citations.

Furthermore, the decision to omit the 0-back condition from the analysis was not justified, especially given the interleaved block design of the N-back task. The 0-back condition serves as an active baseline, controlling for basic visual processing and motor response. Its exclusion raises concerns about whether the reported connectivity patterns reflect working memory-specific processes or more general task-engagement effects.

The study did not explore whether its findings are robust across different cognitive modalities (e.g., face, place, and tool), which can be a compelling extension. This could determine if the high heritability of dFC variance is a general property of working memory or specific to certain modalities.

“only 5% of the connections showed significant differences between MZ and DZ twins. This suggests that static connectivity alone has limited heritability and may not be a strong candidate for an endophenotype”. The term "strong endophenotype" remains poorly defined. The field would benefit from the authors providing a clearer benchmark or context for what proportion of heritable connections would constitute a "strong" candidate. Similarly, the descriptor "moderately heritable" for a 30% heritability estimate is subjective. This should be contextualized by comparing it to heritability estimates for other neurological traits or phenotypes in the literature.

The study identified two distinct dynamic states but provided no further characterization of their functional significance. The analysis would be strengthened by investigating what cognitive processes or neural networks differentiate State 1 from State 2 (e.g., by examining network power or spatial topology) to move beyond a purely statistical description.

The presentation of findings in Figures 2 and 3 lacks clarity. The connectome-wide maps are complex and difficult to interpret. To improve comprehension, these figures can be supplemented with focused illustrations that highlight key working memory regions, making the spatial distribution of heritable connections more accessible.

Minor issues:

Figure 4 was missing error bars, which are essential for assessing the variance and reliability of the presented heritability estimates.

The manuscript frequently used the phrase "We should" which was confusing and gave the impression of a proposed future analysis rather than a completed one. Also, the reporting should be consistently in the past tense for actions that were performed.

Ambiguous terms like "many" should be replaced with specific quantities or percentages (e.g., “According to the heritability analysis using the AE model (Figure 2a) genetic factors influenced many connections across the brain”). Similarly, “We calculate intraclass correlations for twin data and figured out that, ACE model is better describe our data than ADE model. ” The detailed data should be reported.

In results, no need to introduce ACE model again, which was already introduced in methods.

The authors should report specific p values instead of < 0.05

The link to the open data does not seem to work properly.

**Do you want your identity to be public for this peer review?** For information about this choice, including consent withdrawal, please see our Privacy Policy

Reviewer #1: No

Reviewer #2: No

---

## [Author Response · Author response to Decision Letter 1]

1 Dec 2025

See Attached file for details of our point-by-point reply to the reviewers comments.

---

## [Decision Letter · Decision Letter 1]

9 Dec 2025

Genetic Control of Dynamic Brain Network Reconfiguration During Working Memory

PONE-D-25-41450R1

Dear Dr. Daliri,

We’re pleased to inform you that your manuscript has been judged scientifically suitable for publication and will be formally accepted for publication once it meets all outstanding technical requirements.

Kind regards,

Kenji Tanigaki, Ph.D., M.D.

Academic Editor

PLOS One

Additional Editor Comments (optional):

Reviewers' comments:

Reviewer's Responses to Questions

**Comments to the Author**

Reviewer #1: All comments have been addressed

Reviewer #2: All comments have been addressed

2. Is the manuscript technically sound, and do the data support the conclusions?

Reviewer #1: Yes

Reviewer #2: Yes

3. Has the statistical analysis been performed appropriately and rigorously?

Reviewer #1: Yes

Reviewer #2: Yes

4. Have the authors made all data underlying the findings in their manuscript fully available?

Reviewer #1: Yes

Reviewer #2: Yes

5. Is the manuscript presented in an intelligible fashion and written in standard English?

Reviewer #1: Yes

Reviewer #2: Yes

Reviewer #1: The authors have revised their manuscript well. The Discussion section is especially stronger now. I do not have any further comments.

Reviewer #2: (No Response)

**Do you want your identity to be public for this peer review?** For information about this choice, including consent withdrawal, please see our Privacy Policy

Reviewer #1: No

Reviewer #2: No

---

## [Editor Report · Acceptance letter]

PONE-D-25-41450R1

PLOS One

Dear Dr. Daliri,

I'm pleased to inform you that your manuscript has been deemed suitable for publication in PLOS One. Congratulations! Your manuscript is now being handed over to our production team.

Kind regards,

on behalf of

Dr. Kenji Tanigaki

Academic Editor

PLOS One